# Diagnostic Aqueous Humor Proteome Predicts Metastatic Potential in Uveal Melanoma

**DOI:** 10.3390/ijms24076825

**Published:** 2023-04-06

**Authors:** Chen-Ching Peng, Shreya Sirivolu, Sarah Pike, Mary E. Kim, Bibiana Reiser, Hong-Tao Li, Gangning Liang, Liya Xu, Jesse L. Berry

**Affiliations:** 1The Vision Center at Children’s Hospital Los Angeles, Los Angeles, CA 90027, USA; 2USC Roski Eye Institute, Keck School of Medicine, University of Southern California, Los Angeles, CA 90033, USA; 3The Saban Research Institute, Children’s Hospital Los Angeles, Los Angeles, CA 90027, USA; 4Department of Urology, Keck School of Medicine, University of Southern California, Los Angeles, CA 90033, USA; 5Norris Comprehensive Cancer Center, Keck School of Medicine, University of Southern California, Los Angeles, CA 90033, USA

**Keywords:** uveal melanoma, proteomics, aqueous humor, biomarkers, ocular liquid biopsy, ocular cancer

## Abstract

Gene expression profiling (GEP) is clinically validated to stratify the risk of metastasis by assigning uveal melanoma (UM) patients to two highly prognostic molecular classes: class 1 (low metastatic risk) and class 2 (high metastatic risk). However, GEP requires intraocular tumor biopsy, which is limited by small tumor size and tumor heterogeneity; furthermore, there are small risks of retinal hemorrhage, bleeding, or tumor dissemination. Thus, ocular liquid biopsy has emerged as a less-invasive alternative. In this study, we seek to determine the aqueous humor (AH) proteome related to the advanced GEP class 2 using diagnostic AH liquid biopsy specimens. Twenty AH samples were collected from patients with UM, grouped by GEP classes. Protein expression levels of 1472 targets were analyzed, compared between GEP classes, and correlated with clinical features. Significant differentially expressed proteins (DEPs) were subjected to analysis for cellular pathway and upstream regulator identification. The results showed that 45 DEPs detected in the AH could differentiate GEP class 1 and 2 at diagnosis. IL1R and SPRY2 are potential upstream regulators for the 8/45 DEPs that contribute to metastasis-related pathways. AH liquid biopsy offers a new opportunity to determine metastatic potential for patients in the absence of tumor biopsy.

## 1. Introduction

Uveal melanoma (UM) is the most common primary intraocular cancer, characterized by tumors arising from the choroid, iris, and ciliary body [1]. Globe-conserving treatment most commonly entails plaque brachytherapy or proton beam radiotherapy, with enucleation reserved for very large tumors. Even after treatment of the primary tumor, approximately half of all patients with UM will develop metastatic disease, resulting in the need for metastatic surveillance of these patients [2]. However, as half of these patients may not develop metastatic cancer, stratifying the risk of metastasis in these patients can lead to more patient-specific metastatic surveillance recommendations.

Gene expression profiling (GEP) has been shown to yield superior prognostic accuracy in predicting metastasis in UM compared to clinical, histopathologic, and chromosomal features [3,4,5,6], as well as compared to the widely used TNM staging [7]. This method has been validated prospectively and is now being used by clinicians to stratify the risk of metastasis by assigning UM patients to two highly prognostic molecular classes: class 1 (low metastatic risk) and class 2 (high metastatic risk) [3,8]. An optimized test has been developed using a 15-gene expression profile assay that is performed on a microfluidics polymerase chain reaction platform, allowing it to detect small quantities of tumor RNA from both fine needle aspiration samples and surgically resected specimens [3]. This test’s accuracy and high technical reliability has been well established in several retrospective and prospective studies [3,4,9,10,11,12,13,14], and this test is clinically used by physicians to guide metastatic surveillance recommendations [12,15,16].

While tumor biopsies can provide a reliable assessment of the risk of metastatic disease through molecular testing, they are subject to risks of retinal detachment and tumor dissemination [17]. In addition, results from a recent study that consecutively sampled two sites of the same tumor via fine-needle aspiration biopsy (FNAB) show that discordance between GEP results may exist in 11–16% of cases [18]. Their data suggest that a single-site FNAB with a low metastasis result may be misleading in a small minority of patients, as the site of the tumor sampled may be populated by lower-metastatic potential cells than other cell populations in a heterogenous tumor. For these reasons, there is a benefit to establish an ocular liquid biopsy platform for proteomic analysis of UM tumors that can be both less invasive and more representative of the heterogenous subpopulations of tumor cells.

In addition to the limitations of tumor biopsies, genetic analysis does not predict protein expression and posttranslational modification, resulting in an incomplete understanding of the tumor phenotype [19]. Therefore, the analysis of the proteome of AH from UM patients has the potential to improve diagnosis, prognosis, surveillance, and treatment of patients, through providing a more precise characterization of the tumor phenotype. Both the vitreous and aqueous humor have been used as ocular liquid biopsy platforms to study proteomics in UM patients. A recent study of Velez et al. has identified proteomic biomarkers in the vitreous humor of UM patients that were associated with metastatic risk, with their results suggesting mechanisms of tumor proliferation and approaches for adjuvant therapy and metastatic risk surveillance [17]. Another study by Wierenga et al. identified distinct differences between cytokines in different aqueous humor samples of UM patients, which were then allocated into three different prognostic tumor clusters [20]. A recent review by Heiferman et al. describes in further detail the recent advances in proteomics in AH, VH, and tumor of UM patients [19].

Despite the recent advancements in liquid biopsy proteomics in UM patients, there is still not a clinical assay in use that can identify prognostic protein biomarkers at the time of diagnosis. In this study, we are the first to analyze AH from treatment-naïve eyes to identify therapeutic prognostic biomarkers in UM patients, by comparing diagnostic AH samples collected from GEP1 and GEP2 UM eyes. We have previously shown the potential of the AH as an organ-specific liquid biopsy for UM through the presence of tumor-derived cell-free DNA [21]. In this study, we further develop the AH platform through our analysis of how classification of differentially expressed protein biomarkers can compare to the widely used GEP classifications. We seek to correlate the AH protein expression levels with metastatic potential at the diagnostic stage.

## 2. Results

### 2.1. Patient Clinical Characteristics and Demographics

A total of 20 UM AH samples collected at diagnosis, prior to any therapy, from 20 UM patients were analyzed. Patient demographics and clinical characteristics are summarized in Figure 1. BAP1 mutation, preferentially expressed antigen in melanoma (PRAME) status, and GEP class are defined from clinically indicated tumor biopsy when available. A total of twelve (60%) GEP1, five (25%) GEP2, and three (15%) patients without available tumor biopsy were included. All GEP1 tumors were AJCC stage I or II, while 2/5 (40%) of GEP2 tumors were more advanced (*p* = 0.012; Table 1). A significant number of GEP2 tumors were diagnosed at a more advanced clinical tumor stage than GEP1 tumors (*p* = 0.007; Table 1). None of the GEP1 tumors harbor a BAP1 mutation, but 3/5 (60%) of GEP2 tumors had mutant BAP1, a known poor prognostic marker (*p* = 0.018; Table 1).

### 2.2. Target Protein Data Distribution

Using a multiplex proximity extension assay (PEA) platform by Olink, expression levels of 1472 proteins were analyzed from each AH sample (Appendix A). After excluding three low-quality targets, expression levels of 1469 proteins were normalized for variation using internal and external controls, and the generated normalized protein expression (NPX) units were further analyzed. By comparing the NPX data structure, no significant difference in NPX distribution (Figure 2A) between GEP classes was observed, suggesting normalized protein expression units were evenly distributed. Principal components assay (PCA) of the 1469 target NPX values showed two distinct clusters, with 15 AH samples in cluster 1 and 5 AH samples in cluster 2 (Figure 2B). Comparing with FNAB-derived GEP classifications, 11/12 GEP1 cases (91.67%) land in PEA-cluster 1, and 4/5 GEP2 cases (80.00%) land in PEA-cluster 2. All three GEP-unknown cases are included in PEA-cluster 1 (Figure 2B).

### 2.3. Differentially Expression Proteins between GEP2 and GEP1 AH Samples

To identify the AH differentially expressed proteins (DEPs) related to GEP classes, NPX values of GEP2 samples were compared with GEP1 samples by setting the threshold that *p* value (Mann–Whitney U test) is smaller than 0.01 and that log2 (fold change) is greater than 1 or smaller than −1. Among the identified 45 DEPs, 31/45 (68.9%) and 14/45 (31.1%) were found to be upregulated and downregulated in GEP2, respectively (Figure 3A). To analyze the discrimination power of the DEPs, we further perform an unsupervised clustering of all 20 AH samples using the 45 DEPs. It was found that 5/5 GEP2 samples formed a distinct cluster on the left side of the heatmap, with only 1 GEP1 case (UM_021) clustered with GEP2 samples (Figure 3B). In addition, 9/12 GEP1 and 3/3 GEP NA samples formed the right side of the heatmap, suggesting all 3 GEP NA cases (UM_010, 015 and 020) may be GEP1 diseases (Figure 3B). Two GEP1 cases (UM_005 and 017) clustered closer to the left side, suggesting they might be in transitional stages from benign diseases to more advanced diseases.

### 2.4. Biological Significance of the DEPs

To determine the biological relevance of the 45 DEPs, pathway analysis was conducted to identify the important cellular function these DEPs may contribute. After subjecting the 45 DEPs to Qiagen Ingenuity Pathway Analysis (IPA), we identified pathways related to inflammatory response, cellular growth (cellular growth and proliferation and cell death and survival)- and cell motility (cellular movement and epithelial-to-mesenchymal transition (EMT))-related with a significant z-score (Figure 4A). We next performed an upstream regulator analysis (URA) to identify the potential upstream target that has been experimentally verified to affect the 45 DEP expression. The most significant upstream regulators of the 45 DEPs were TNF, FGF2, IL-1 receptor, MYD88, and SPRY2 (Figure 4B, −log *p* value > 5). In the PEA platform, upregulation of IL-1 receptor 2 (*p* = 0.019, Figure 4C) and downregulation of SPRY2 (*p* = 0.052, Figure 4C) were found to be concordant with the URA z-score (Figure 4B), while MYD88 was not in our 1472 protein panel. The downstream targets of IL-1 receptor and SPRY2 within the 45 DEPs are listed in Figure 4D. Of note, co-targeting of IGFBP3 and PLAUR by both IL-1 receptor and SPRY2 (Figure 4D) may lead to the subsequent activation of their protein expression in GEP2 samples (Figure 4E).

## 3. Discussion

While gene expression profiling is a clinically validated and widely established method to stratify the risk of metastasis in UM patients, it requires intraocular tumor biopsy. This is limited by small tumor size, tumor heterogeneity, as well as the risk of retinal hemorrhage, bleeding, and tumor dissemination, albeit quite rare. We hypothesized that with minimally invasive aqueous humor extraction from UM eyes, we are able to provide metastatic risk stratification similar to gene expression profile (GEP) classification by FNAB. The first step to achieve this long-term goal is to identify if there are differential patterns of molecules existing in diagnostic UM AH samples. Due to the low abundance and highly fragmented status of nucleic acid in pre-radiation AH samples, we aimed to instead determine the differential expression of proteins (DEPs) between GEP classes from the AH of 20 treatment-naïve eyes in this study.

The use of proximity extension assay (PEA)-derived multiplexed technology enabled the comparison of 1472 protein targets in a small volume of AH samples. A total of 45 differentially expressed proteins were identified when comparing quantities of these 1472 targets in the AH from 12 GEP1 patients and 5 GEP2 patients, which were then correlated with clinical features. After multiple-hypothesis adjustment using the Bonferroni correction, TNFRSF12A was identified to be significantly differentially expressed. Due to our small explorative sample size, the rest of the 45 DEPs show trends to be associated with the advanced GEP class (GEP2). While TNFRSF12A is the most impactful and significant predictor, other DEPs on the 45 DEP signature panel are required to identify all cases as GEP1 vs. GEP2. Thus, there is likely value in including the full signature panel to capture the protein expression, and thus the metastatic potential, of each case. This will require future prospective evaluation in a larger cohort of patients. The three samples for which the GEP was unknown, due to sample availability, are presumed likely to be GEP1 due to their 45 DEP-based classification. Notably, while these tumors were too small to biopsy, they were able to be classified into a presumed GEP class based on their protein expression from this assay. A principal component analysis (PCA) of all 1469 protein targets (Appendix A) between 3 congenital contacts (CAT), 4 congenital glaucoma (GLC), and 20 UM AH samples is shown in Appendix A. This analysis demonstrates that all UM samples are grouped separately from CAT and GLC, suggesting that we may be able to differentiate UM from these diseases using the AH proteome; however, this will require future larger studies and multiple disease entities for validation. It should be noted, however, that the intent of the 45-DEP signature is meant to be prognostic within UM and not diagnostic for UM.

Among these 17 cases with known GEP class, all GEP class 2 cases were identified by this DEP clustering, while 3 GEP class 1 cases (UM_005, 017, and 021) were not concordant with their DEP clustering. UM_021 (GEP1), which sits in the middle of other GEP2 cases, is stage T3 with an 8.28 mm tumor size in height. It has been shown that the incorporation of tumor size improves the prognostic ability of GEP classifications in patients with posterior UM; it is possible that there are clonal expression differences in this large tumor so that the AH analysis identifies this large tumor as class 2, while the tumor biopsy identified it as class 1 [22]. Genomic analysis can be performed via the AH in addition to protein expression analysis to identify other markers of metastatic risk. Somatic copy number alteration profiling for UM_005 identified a gain in 8q (Appendix A), which is known to correlate with higher rates of metastasis and the risk of mortality [22,23]. SCNA analysis also revealed a gain of chromosome 6p. The combined gain of chromosome 8q and 6p places the patient in The Cancer Genome Atlas (TCGA) class B, again suggesting an increased risk of metastasis [24]. Further, the patient is PRAME positive, suggesting this patient has three risk factors for poor prognosis despite being GEP1 from tumor biopsy. There were no positive UM_SCNAs identified in UM_017 due to the low quantity of DNA available for genomic analyses. While these represent a small number of cases, it suggests that AH DEP analysis, especially when combined with genomic analysis, may have increased utility in capturing molecular biomarkers indicating poor prognosis.

The cellular functions of these DEPs were found to be associated with cell growth and motility. The upstream regulator analysis showed that the significant upstream regulators of the 45 DEPs were TNF, FGF2, IL-1 receptor, MYD88, and SPRY2. Both the upregulation of IL-1 and the downregulation of SPRY2 were also shown in the Olink PEA-derived platform. The upregulation of IL-1 has been shown in other cancers, including breast, colon, head and neck, lung, pancreas, and melanomas, with patients with high levels of IL-1 having generally poor prognosis [25]. The downregulation of SPRY2 has been shown in chronic lymphocytic leukemia, non-small cell lung cancer, hepatocellular carcinoma, breast cancer, and prostate cancers [26], highlighting its tumor suppressor function.

Prior studies of UM patients by Velez et al. and Wierenga et al., which analyzed the vitreous and aqueous humor, respectively, have identified additional prognostic protein biomarkers. Velez’s work identified protein signatures in the vitreous humor that correlated with GEP and PRAME. Although we found similar elevated trends of their targets (SCFR/c-KIT, HGFR/c-MET, and SIGLEC6) in the AH of GEP2 eyes, no statistically significant difference was observed (Appendix A). It might be due to the fact that they compared AJCC stage III–IV vs. I–II tumors at validation, while we only had two AJCC stage III tumors in our cohort. While their approach, using the vitreous humor, and ours, using the aqueous humor, both aim to provide prognostic biomarkers via a minimally invasive method for real-time intraocular assessment of UM, our method has the potential benefit of being even less invasive. In contrast, after analyzing 92 cytokines in the AH from 84 enucleated UM eyes showing distinct patterns, Wierenga et al. allocated these samples into three different prognostic tumor clusters that were analyzing metastatic death, as opposed to intraocular metastatic risk. Still, high concordance was observed on overlapped targets in our analyses (Appendix A), suggesting we could converge our efforts with their PEA-based proteomic dataset for future combined ocular and life mortality analysis.

Gene expression profiling (GEP) by Castle Biosciences is based off a 15-gene panel. The twelve genes of interest are: CDH1, ECM1, EIF1B, FXR1, HTR2B, ID2, LMCD1, LTA4H, MTUS1, RAB31, ROBO1, and SATB1. The three control genes are MRPS21, RBM23, and SAP130. PRAME expression is an independently validated marker. While none of these genes were included in the 45-DEP signature panel, of these 13 (12 from GEP +1 for PRAME) genes of interests in UM, we include protein expression from CDH1, LTA4H, and ROBO1, which are present in the larger dataset of 1469 proteins. Of these three, we observed a significant (*p* < 0.05) difference in expression in CDH1 between GEP1 and GEP2 (Appendix A), but not LTA4H or ROBO1.

The identification of biomarkers, such as SPRY2 and IL-1 from our upstream regulator analysis, can suggest mechanisms of tumor proliferation and approaches for adjuvant therapy, due to the potential of such proteins to become eventual targets for drug repositioning. As the use of different assays on different liquid biopsy origins (AH or VH) has provided distinct prognostic biomarkers, there is a potential for additional studies with similar aims to continue identifying more biomarkers, which may increase our likelihood of identifying a suitable therapeutic protein target for these patients. Moreover, in vivo proteomic profiling from AH may improve our understanding of the biological mechanisms for tumor spread and suggest rational approaches for adjuvant therapy and metastatic risk surveillance.

There are multiple limitations to this early-stage study, including the small sample number, high number of readouts, and potential intra-tumor heterogeneity. A future prospective analysis will help in verifying these prognostic biomarkers. Despite these limitations, aqueous humor liquid biopsies may be a desirable alternative or a complement to tissue biopsy. An AH liquid biopsy is minimally invasive, and it can be performed regardless of tumor size, which impacts the technical feasibility of a biopsy, and unlike a biopsy, AH extraction is easily repeated, which is conducive to longitudinal treatment monitoring. Finally, multi-omics approaches are only increasing, and AH analysis may provide critical details that complement other analyses performed on biopsied tissue. Most impactful is the ability to analyze the protein expression in the aqueous humor, which has the potential to provide critical information of metastatic potential in cases in which the tumor is too small to biopsy directly. This is illustrated by the three cases included in our analysis, which were too small to biopsy (posterior tumors with <2.5 mm height), yet they yielded meaningful proteomic signals in AH analysis.

The results of this study suggest that proteomic analysis of AH has a potential for clinical utility as a surrogate to tumor biopsy-based staging, such as GEP classification, for determining metastatic potential.

## 4. Materials and Methods

This investigation was a case series study at a tertiary care hospital (Roski Eye Institute, University of Southern California, Los Angeles, CA, USA). Samples were taken between August 2020 and May 2021.

### 4.1. Patient Clinical Characteristics and Demographics

This study included a convenience sample of 20 UM patients at the Roski Eye Institute, University of Southern California, from whom written informed consent for an AH sample was obtained. All samples consisted of ~0.1 mL of AH extracted via clear cornea paracentesis at the end of surgery for brachytherapy plaque placement. We included 20 AH samples from 20 UM eyes. Proteomics results were coded and maintained separately from clinical data and thus did not alter patient treatment for all participants.

### 4.2. Specimen Collection and Storage

A clear corneal paracentesis with a 30-gauge needle was performed to extract ~0.1 mL of AH from UM eyes during clinically indicated surgery to treat UM. The extraction method has been described in detail and published previously by our group for specimen collection from retinoblastoma eyes [27]. Briefly, needles only entered the anterior chamber via the clear cornea at the limbus and did not make contact with the iris, lens, vitreous, or UM tumor. Samples were stored on dry ice immediately and transferred to −80 °C within hours of extraction. Routine FNAB with either a 25- or 27-gauge needle was conducted on 16 patients for mutational analysis and 17 patients for GEP and PRAME status, which was performed at Castle Biosciences (Phoenix, AZ, USA).

### 4.3. Quantification of Protein Expression by Proximity Extension Assay (PEA)

AH samples were analyzed for 1472 proteins using the Olink Explore PEA assay (Olink^®^ Proteomics, Uppsala, Sweden). Briefly, protein-specific antibodies tagged with DNA sequences were used, followed by a NGS PEA, which gave abundance levels for each protein measured as NPX values (normalized protein expression) on a log2 scale. For details and full list of proteins, see Appendix A. The PEA assay was chosen over other protein assays due to several factors. Other approaches for protein detection, such as mass spectroscopy and ELISA, have some disadvantages when measuring proteins of low abundance: ELISA is not scalable to measure >90 proteins at a time per sample, and mass spectroscopy favors highly abundant proteins. With PEA, it is possible to measure proteins of low abundance with high sensitivity and specificity while enabling high throughput using a minimal amount of sample. This makes the assay ideal for measuring a high range of proteins in a large number of samples.

### 4.4. Gene Ontology Analysis and Annotations

QIAGEN’s Ingenuity^®^ Pathway Analysis (IPA^®^, QIAGEN, Hilden, Germany) software (web version, www.qiagen.com/ingenuity, accessed on 1 October 2022) was used for functional annotation pathway analysis and upstream regulatory analysis (URA) of the 45 DEPs. Activation z-scores and the corresponding *p* values were generated by IPA^®^. Z-score predicts the activation (positive value) or inhibition (negative value) of a canonical pathway/upstream regulator. Absolute z-score values ≥ 2 are considered significant.

### 4.5. Statistical Analysis

Continuous variables were represented as the mean ± standard deviation (S.D.), and non-normally distributed variables were compared by the Mann–Whitney U test. Bonferroni’s correction was used for multiple hypothesis adjustment. Categorical variables were compared using the Fisher’s exact test or linear-by-linear association test, as indicated in Table 1. All statistical tests were two-tailed, and *p* < 0.05 was considered statistically significant. p-values are represented as: *, *p* < 0.05; **, *p* < 0.01; ***, *p* < 0.001; ns, non-significant. All statistical analyses and plots were conducted using the Prism 8 (GraphPad, San Diego, CA, USA).

### 4.6. Analysis of Somatic Copy Number Alterations (SCNAs)

Analysis of the cfDNA from AH samples was previously outlined in depth and based on established methods of SCNA analysis [27,28]. Briefly, isolated cfDNA was constructed into a whole-genome library (QIAGEN, Hilden, Germany) followed by shallow whole-genome sequencing for copy number profiling. SCNAs were considered to be present at 20% deflection from a baseline human genome, consistent with established liquid biopsy analyses [29].

## 5. Conclusions

Analyzing the proteome in the aqueous humor offers a new opportunity to determine uveal melanoma metastatic potential, with notable clinical utility for small-sized tumors that cannot be biopsied.

## 6. Patents

Authors Berry and Xu have filed a patent application entitled: Aqueous humor cell-free DNA for diagnostic and Prognostic evaluation of Ophthalmic Disease.

## Figures and Tables

**Figure 1 ijms-24-06825-f001:**
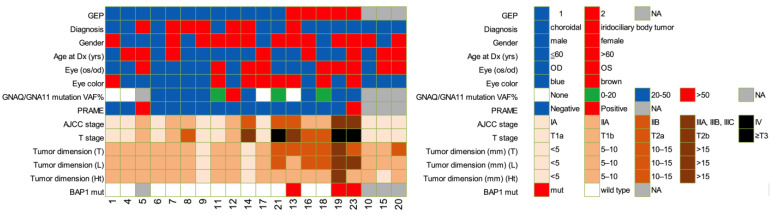
Patient demographic data for the entire cohort. Abbreviations: GEP status, gene expression profile; Dx, diagnosis; PRAME, preferentially expressed antigen in melanoma; AJCC, American Joint Committee on Cancer (AJCC). T stage, tumor stage; Tumor dimension (mm) T, width, L, length, Ht, height; Mut, mutation.

**Figure 2 ijms-24-06825-f002:**
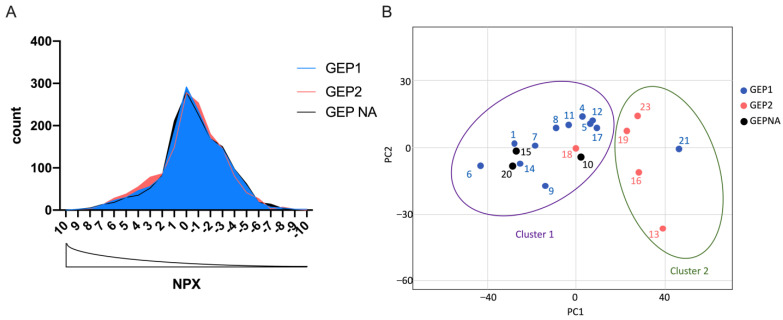
Normalized protein expression (NPX) distribution from 20 uveal melanoma (UM) aqueous humor (AH) samples. (**A**) Protein count versus NPX distribution and (**B**) Principal component analysis of 1469 proteins in 20 UM AH samples. Gene expression profile 1 (GEP1): low likelihood of metastasis; gene expression profile 2 (GEP2): high likelihood of metastasis; gene expression profile unknown (GEP NA): limited access of tumor biopsy for performing GEP analysis.

**Figure 3 ijms-24-06825-f003:**
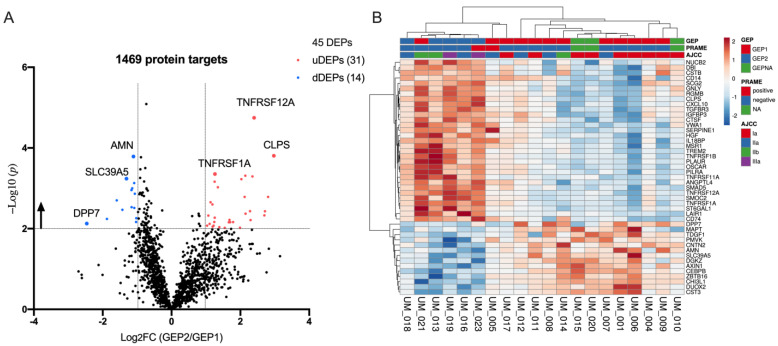
Differentially expressed aqueous humor (AH) proteins in gene expression profile 2 (GEP2) versus gene expression profile 1 (GEP1) samples. (**A**) Differentially expressed proteins (DEPs) of AH in GEP2 versus GEP1 samples are defined as –log (*p*) greater than 2 (*p* value < 0.01) and |log2 fold change (FC)| greater than 1 (fold change >2 or <0.5, GEP2 over GEP1). Red dots denote 31 upregulated DEPs (uDEPs), and blue dots are 14 downregulated DEPs (dDEPs). Black dots are the rest of the 1424 proteins in this panel. The *p* value was calculated by the Mann–Whitney U test. (**B**) Unsupervised clustering of 20 uveal melanoma (UM) AH samples using the 45 DEPs. GEP classes, PRAME status, AJCC stages, and the color chart representing expression levels for each protein are indicated. Gene expression profile 1 (GEP1): low likelihood of metastasis; gene expression profile 2 (GEP2): high likelihood of metastasis; gene expression profile unknown (GEP NA): limited access of tumor biopsy for performing GEP analysis. PRAME, preferentially expressed antigen in melanoma; AJCC, American Joint Committee on Cancer (AJCC).

**Figure 4 ijms-24-06825-f004:**
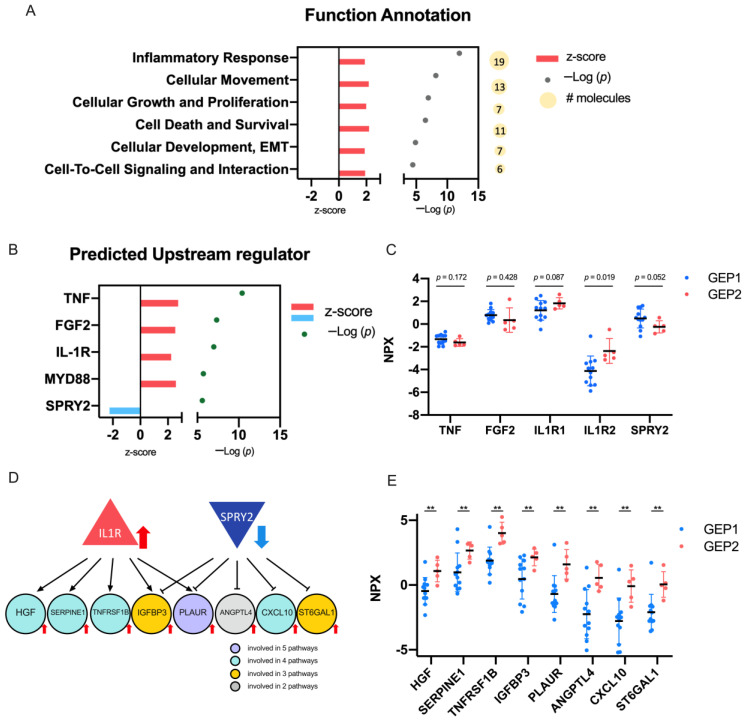
Biological relevance of the differentially expressed proteins (DEPs). (**A**) Qiagen Ingenuity Pathway Analysis (IPA) function annotation analysis of the 45 DEPs. Z-score, –Log *p* values and number of DEPs (yellow circles) contributing to each pathway are indicated. (**B**) Upstream regulator analysis of the 45 DEPs. Top 5 significant upstream regulators are plotted. Increased (red) and decreased (blue) z-score and –Log *p* values are shown. (**C**) Protein expression levels of potential upstream regulators, namely, TNF, FGF2, IL-1 receptor 1 and 2, and SPRY2, in the proximity extension assay (PEA) platform. (**D**) Potential downstream targets of IL-1 receptor and SPRY2 in 45 DEPs. Color of each circle represents the number of cellular pathways the targets are involved in. Red arrows indicates upregulation, blue arrows represent downregulation. (**E**) Protein expression levels of IL-1 receptor and SPRY2 targets, HGF, SERPINE1, TNFRSF1B, IGFBP3, PLAUR, ANGPTL4, CXCL10, and ST6GAL1. *p* values were calculated by the Mann–Whitney U test. Gene expression profile 1 (GEP1): low likelihood of metastasis; gene expression profile 2 (GEP2): high likelihood of metastasis; gene expression profile unknown (GEP NA): limited access of tumor biopsy for performing GEP analysis. **, *p* < 0.01.

**Table 1 ijms-24-06825-t001:** Univariate comparison of clinical characteristics between GEP1 and GEP2 UM patients.

Characteristic	GEP1, n = 12	GEP2, n = 5	*p*
Sex (Fisher), n (%)			0.600
Females	7 (58.3)	4 (80)	
Males	5 (41.7)	1 (20)	
Eye (Fisher), n (%)			0.620
OD	7 (58.3)	2 (40.0)	
OS	5 (41.7)	3 (60.0)	
Age at diagnosis, mean (± SD) (MWU)	53.3 (15.5)	62.8 (12.7)	0.183
Eye Color (Fisher), n (%)			0.620
Light (blue, gray, green, hazel)	7 (58.3)	2 (40.0)	
Dark (brown)	5 (41.7)	3 (60.0)	
Anterior Involvement (Iris and/or ciliary) (Fisher), n (%)			0.338
Yes	6 (50.0)	1 (20.0)	
No	6 (50.0)	4 (80.0)	
AJCC Stage (Linear-by-Linear association), n (%)			0.012
I	6 (50.0)	0 (0)	
IIA	4 (33.3)	2 (40.0)	
IIB	2 (16.7)	1 (20.0)	
IIIA, IIIB, IIIC	0 (0)	2 (40.0)	
IV	0 (0)	0 (0)	
PRAME Status, known in 17 cases (Fisher), n (%)			0.515
Negative	11 (91.7)	4 (80.0)	
Positive	1 (8.3)	1 (20.0)	
GEP Class, known in 17 cases (Fisher), n (%)			<0.001
1	12 (100.0)	0 (0)	
2	0 (0)	5 (80.0)	
Tumor Stage (Linear-by-Linear association), n (%)			0.007
T1	9 (75.0)	0 (0)	
T2	2 (16.7)	3 (60.0)	
T3	1 (8.3)	1 (20.0)	
T4	0 (0)	1 (20.0)	
BAP1 mutation status, known in 16 cases (Fisher), n (%)			0.018
Mutation	0 (0)	3 (60.0)	
Wild type	11 (100.0)	2 (40.0)	

AJCC, American Joint Committee in Cancer; Fisher, Fisher’s exact test; GEP, gene expression profile; MWU, Mann-Whitney U test; PRAME, preferentially expressed antigen in melanoma; SD, standard deviation.

## Data Availability

De-identified original dataset has been uploaded as Appendix A.

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
