# Peer review of "Diagnostic Aqueous Humor Proteome Predicts Metastatic Potential in Uveal Melanoma"

_ijms, 2023, doi:10.3390/ijms24076825_

Round 1

Reviewer 1 Report

Peng and colleagues describe an interesting study that uses aqueous humor liquid biopsies to investigate the proteome of patients affected by uveal melanoma, and to implicate protein expression with high or low metastatic risk classes, identifiable with gene expression profiling. The authors collected and profiled 20 samples and uncovered 45 proteins and their regulators that were differentially expressed between the two classes. They thus conclude that aqueous humor proteome can be used to determine the metastatic potential of uveal melanomas. 

This study provides a significant advancement to the field and is clear, well written and structured. The results are solid and presented in the context of the current literature. However, addressing the few points that I outline below would further improve the soundness of the results and the clarity of the article.

Major points

1) As the authors acknowledge in the Discussion, a limitation of this study is the small number of samples from which the conclusions are drawn. As a result, it would be important to validate the findings on an external, larger-scale dataset, for example TCGA-UVM (80 samples). TCGA-UVM clinical table provides information regarding the localized/metastatic nature of each tumor (columns "metastatic_site", "pathologic_m"), along with multiple omics profilings. Protein data is unfortunately limited (12 samples only), but mRNA is available for all 80 cases. Assuming that on average the RNA expression and protein levels should correlate (although it might not always be the case for several reasons), I suggest that the authors check whether at least some of their differentially expressed proteins between GEP1 and 2 overlap the differentially expressed genes in TCGA between localized and metastatic TCGA-UVM samples. The authors'-derived 45-proteins signature could even be tested for association with survival and stage in the TCGA dataset.

2) A natural question would be if the 15 genes of the RNA signature translate into proteins that are in turn differentially expressed between GEP1 and GEP2. I am aware that not all of them would be found measured in the protein assay of this study, but I could spot a few of them (CDH1, LTA4H, ROBO1). I noticed that the protein levels of these is reported in Figure A4, but this does not seem to be referenced in the text. I suggest at least to mention and discuss this comparison.

Minor points

3) It would be helpful to have a descriptions of the columns of Supplementary Table 1 in the Appendix (e.g. what is LOD?). Additionally, among the samples there are some that are not referenced in the text (those starting with "CAT_" and "GLC_")

4) Could you retrieve the RNA levels that were used to assign the GEP1/2 category for each of these 20 cases? If so, would it be possible to directly correlate the levels of CDH1, LTA4H and ROBO1 in the RNA and protein expression?

5) How were somatic copy numbers measured in patient UM_005 (Figure A2)? Which technology and which data analysis pipeline were used? Please add a small paragraph in the Methods section describing this

Reviewer 2 Report

Diagnostic aqueous humor proteome predicts metastatic potential in uveal melanoma

In the study presented the authors describe that AH liquid biopsy offers a new opportunity to determine metastatic potential for patients in the absence of tumor biopsy. The authors analyzed 20 AH from uveal melanoma patients by Olink Explore PEA assay and identified differentially expressed AH proteins in two groups of UM patients. Different analyses identified pathways and predicted upstream regulators. In general, the results are very interesting and helpful to establish a noninvasive diagnostic approach for UM patients for the prognosis of metastatic risk. Nevertheless, the results are preliminary (small cohort) and are not validated by any other method.

1)    The authors should validate their results in order to be sure if the identified protein targets and predicted upstream regulators are differentially regulated in the two groups of UM. They have tumor biopsies from some patients to validate the expression in tumor material (RNA or protein level) and correlate the results with AH. In addition, the authors could analyze the identified targets in another larger cohort of retrospective cases via paraffin sections of UM tumors and correlate the results with the clinical parameters.

2)    It would be good to know if the expressed proteins are specific for UM. Therefore, the authors could analyze AH from other tumor entities (for example RB) or non-tumor patients (may not be available) or cell culture supernatant from UM cell lines in order to show that the proteins are potential biomarkers for UM.

Round 2

Reviewer 1 Report

I thank the authors for addressing the points I raised. 

However, there is another point that should be clarified to ensure the statistical soundness of the findings. In section 2.3, differentially expressed proteins are identified by filtering on the nominal p-values. I had assumed that these p-values were also corrected for multiple hypothesis testing, as it is the standard procedure, but it does not seem to be the case. It is relevant here since 1469 proteins are assayed (and thus 1469 hypotheses are tested). The authors should use a multiple hypothesis testing correction procedure such as Benjamini-Hochberg or Bonferroni to adjust the nominal p-values. Proteins below a certain threshold for adjusted p-values can be deemed significantly differentially expressed, the others would only show a trend.

If not all the 45 DEP proteins used later in the pathway analysis have significant adjusted p-values, the authors should at least acknowledge in the main text that the p-vales are no longer significant after multiple hypothesis correction.

Author Response

Response: Thank you for this very important comment. We entirely agree that multiple hypothesis correction is an ideal way to test the significance among this 1469 protein data and this was an oversight on our part. We have edited the text accordingly:

 in line 125 to specify the test we used in section 2.3:

 Mann-Whitney U test.”  

 and 195-203 in 3. Discussion section to add

 After multiple hypothesis adjustment using the Bonferroni's correction, 2 out of the total protein targets (RET and TNFRSF12A) were identified to be statistically significant in their differential expression. Due to our small explorative sample size, the rest of the DEPs show trends towards significance. While RET and TNFRSF12A are the most impactful and significant predictors, other DEPs on the 45 DEP signature panel are required to identify all cases as GEP1 vs GEP2. Thus, there is likely value in including the full signature panel to capture the aqueous humor protein expression, and thus the metastatic potential, of each case. This will require future prospective evaluation in a larger cohort of patients.”

 And

336-337 in section 4.5 to add

”Bonferroni’s correction was used for multiple hypothesis adjustment.”

Reviewer 2 Report

The authors answered my questions.

Author Response

Response: Thank you.

Round 3

Reviewer 1 Report

Thank you for addressing my last point. 

Just one important note: the genes that you report passing the Bonferroni correction in the Discussion are TNFRS12A (indeed the most DEP from Figure 3A) and RET. However, RET does not appear to be differentially expressed (it is not in the heatmap of Figure 3B and I don't find it as differentially expressed in my analyses).

I would thus suggest to double check that RET is correct and not a typo (maybe RETN?)

Other than this, congratulations for this study and for making the data available to other researchers.

Author Response

Response:
Thank you for the very dedicated review. We agree that introducing RET here is confusing.

Both RET and RETN are proteins in the larger list but not in our signature panel (RET (uniport ID: P07949) and RETN (uniport ID: Q9HD89)

While the adjusted p value for RET after Bonferroni’s correction is significant, the overall fold change of RET between GEP1 and GEP2 is lower than our selection criteria for DEPs: Log2 (fold change) >1 or < -1. RET is a significant differentially expressed protein but not with a high enough fold change to be included in our 45-DEP list. We also did FDR analysis, with a Q of 10%. As such 10 of our 45 DEP signature panel was significant but also others with lower fold change.

All of this will require a larger validation study, preferably with multiple sites, and we are actively engaged in this.

Thus, to not introduce confusion, we have removed RET from the discussion. 

The paragraph in discussion from line 195-201 now reads:
 “After multiple hypothesis adjustment using the Bonferroni's correction, TNFRSF12A was identified to be significantly differentially expressed. Due to our small explorative sample size, the rest of the 45 DEPs show trends to be associated with the advanced GEP class (GEP2). While TNFRSF12A is the most impactful and significant predictor, other DEPs on the 45 DEP signature panel are required to identify all cases as GEP1 vs GEP2. Thus, there is likely value in including the full signature panel to capture the protein expression, and thus the metastatic potential, of each case.”
